# The nanoCUT&RUN technique visualizes telomeric chromatin in Drosophila

Tao Chen[1,2☯], Xiaolu Wei[3☯], Cécile Courret[4☯], Min Cui[1,2], Lin Cheng[1,2], Jing Wu[2], Kami Ahmad[5]*, Amanda M. Larracuente[3,4]*, Yikang S. Rong[2]*

1 Sun Yat-sen University, Guangzhou, China, 2 University of South China, Hengyang, China, 3 Department of Biomedical Genetics, University of Rochester Medical Center, Rochester, New York, United States of America, 4 Department of Biology, University of Rochester, Rochester, New York, United States of America, 5 Fred Hutchinson Cancer Research Center, Seattle, Washington, United States of America

☯ These authors contributed equally to this work.
* kahmad@fredhutch.org (KA); alarracu@bio.rochester.edu (AML); zdqr03@yahoo.com (YSR)

## Abstract

Advances in genomic technology led to a more focused pattern for the distribution of chromosomal proteins and a better understanding of their functions. The recent development of the CUT&RUN technique marks one of the important such advances. Here we develop a modified CUT&RUN technique that we termed nanoCUT&RUN, in which a high affinity nanobody to GFP is used to bring micrococcal nuclease to the binding sites of GFP-tagged chromatin proteins. Subsequent activation of the nuclease cleaves the chromatin, and sequencing of released DNA identifies binding sites. We show that nanoCUT&RUN efficiently produces high quality data for the TRL transcription factor in Drosophila embryos, and distinguishes binding sites specific between two TRL isoforms. We further show that nanoCUT&RUN dissects the distributions of the HipHop and HOAP telomere capping proteins, and uncovers unexpected binding of telomeric proteins at centromeres. nanoCUT&RUN can be readily applied to any system in which a chromatin protein of interest, or its isoforms, carries the GFP tag.

## Author summary

The method of chromatin immunoprecipitation followed by genomic sequencing (ChIP-seq) has been employed to study the distribution of chromatin binding proteins genome-wide. Such studies have greatly enhanced our understanding of the function of the target proteins. However, the uses of chemical crosslinking combined with the procedure of antibody-mediated precipitation of the protein-DNA complex have limited the efficiency of ChIP-seq. The recently developed CUT&RUN method has greatly improved that efficiency. We here developed the "nanoCUT&RUN" extension of CUT&RUN, which can be readily applied to any target protein with a GFP tag. Using nanoCUT&RUN, we profiled the HipHop and HOAP proteins that protect telomeric chromatin in Drosophila. We uncovered sequence-independent binding of both proteins predominantly to telomeres. Interestingly, HipHop binding can also be detected at centromeric chromatin suggestive of a novel function of a telomere capping protein.

**Data Availability Statement:** The plasmid for nGFPMNase expression has been deposited to Addgene (ID: 187826). Sequence reads are available on the NCBI short reads archive with the accession number PRJNA723550 (https://www.

ncbi.nlm.nih.gov/bioproject/PRJNA723550), and code and data files for figures are https://github.com/LarracuenteLab/nanoCUTandRUN.github.repo. Data underlying figures are deposited in the Dryad Digital Repository under https://doi.org/10.5061/dryad.zcrjdfngc. The authors affirm that all data necessary for confirming the conclusions of the article are present within the article, figures, and tables.

**Funding:** The research has been supported by grants from the National Key R&D Program of China (2018YFA0107000) and National Natural Science Foundation of China (31371364) to YSR, National Institutes of Health R01HG010492 to KA and National Institutes of Health R35GM119515 and National Science Foundation MCB 1844693 to AML, and an Agnes M. Messersmith and George Messersmith Dissertation Fellowship to XW. The funders had no role in study design, data collection and analysis, decision to publish, or preparation of the manuscript.

**Competing interests:** The authors have declared that no competing interests exist.

## Introduction

Telomeres protect the natural ends of linear chromosomes from being recognized as DNA breaks. In most organisms studied, chromosome ends are elongated by the enzyme telomerase using an RNA template. Telomerase-synthesized DNA repeats serve as binding sites for sequence specific binding proteins essential for end protection (reviewed in [1]). However, in many organisms such as the model Drosophila and particularly Dipteran insects, either telomerase is missing, or it is missing a conserved domain necessary for high processivity [2–4]. In these organisms, retrotransposons or other sequences populate the ends of chromosomes. Despite possessing vastly different end sequences, at least some of these "telomerase-less" systems rely on a reverse transcription-based mechanism for end elongation. In addition, telomere-specific binding proteins have been identified, at least in Drosophila, that serve similar end protection functions as the sequence-specific binding proteins in the telomerase-containing systems [e.g., 5–8]. Among telomere-binding proteins is a class of so-called capping proteins that, when missing, renders chromosomes susceptible to end-to-end fusion. Drosophila capping proteins have been collectively called "Terminin" [7], similar to the concept of "Shelterin" proposed for telomerase-maintained systems [9]. How capping proteins protect chromosome ends remains one of the major research topics in the field of telomere and genome maintenance.

In Drosophila, how capping proteins are recruited to telomeres remains obscure. Unlike the sequence-specific binding of Shelterin components in mammals, Drosophila Terminin proteins are believed to be sequence-independent. It has been known for over 30 years that a Drosophila chromosome can be stably inherited for generations without the presence of telomeric retrotransposons [10 and references therein]. More recent genomic analyses uncovered surprisingly frequent events in which the entire transposon array is lost from one or more telomeres in natural populations [11,12]. Some Drosophila species appear to have lost the telomeric retrotransposons [13]. Moreover, we showed that a DNA fragment from the non-telomeric *white* locus is occupied by the HipHop and HOAP capping proteins only when the gene is situated at the very end of a chromosome [6]. These results suggest that capping protein binding does not require a sequence component from Drosophila telomeres. However, the natural binding partners of Drosophila capping proteins remain the three classes of non-LTR retrotransposons that are specifically enriched at chromosome ends. Therefore, one cannot rule out the hypothesis that there are "preferred" binding sites on the transposons that the capping proteins rely on for proper localization. In addition, physical interaction between these transposons and the proteins that bind them have been proposed to drive the rapid evolution of Drosophila telomeres [14,15]. Therefore, characterization of telomeric protein binding on endogenous chromosomes in a telomerase-less system are important for a better understanding of the biology and evolution of telomeres and their functions. Here we profile these binding sites for the first time in the *Drosophila melanogaster* model. We chose the recently developed "CUT&RUN" technique [16], but with our own modifications.

In 2004, Laemmli and colleagues [17] developed the Chromatin ImmunoCleavage (ChIC) technique in which the Micrococcal Nuclease (MNase) is brought to the vicinity of a target protein by an interaction between Protein A and the target bound antibody. Bound MNase, which had been purified from bacteria as a fusion protein with Protein A, is activated by the addition of calcium and cleaves DNA around the site of target protein binding. This principle of targeted cleavage was further explored by Skene and Henikoff [16] to achieve efficient separation of the cleaved fragments from the bulk of uncut chromatin, and when combined with second generation sequencing led to the Cleavage Under Targets and Release Using Nuclease (CUT&RUN) method.

In CUT&RUN, similar to other ChIP methods, the principal target specificity is determined by antigen-antibody interactions. Therefore, antibodies might have to be developed for every target protein, and in special cases for every isoform of interest from loci encoding multiple ones, such as the *mod(mdg4)* locus in Drosophila that encodes as many as 31 different isoforms [flybase.org]. These limitations could be overcome by using epitope tagging so that a single anti-Tag antibody could be used to profile different targets or isoforms of a single target. An added advantage of using a common anti-Tag antibody is that profiles of different targets/isoforms could be directly compared as long as their relative expression levels have been taken into consideration.

In our studies of Drosophila chromatin proteins, we often employed tagging with a Green Fluorescent Protein (GFP) [e.g., 18–21] so that live imaging of the target proteins could be achieved as long as the GFP-tagged target had been proven functional in genetic rescuing experiments. In theory, an anti-GFP antibody could be used to conduct CUT&RUN profiling of any such target. However, our collective experiences with monoclonal anti-GFP antibodies have been unsatisfactory. A nanobody is an antigen-binding fragment derived from the variable domain of a heavy chain only antibody produced in species such as the Camelids. Since single-domain anti-GFP nanobodies offer consistent performance in IP-related experiments [e.g., 22,23], and can be readily purified from bacteria, we were prompted to develop a method generally applicable to profiling GFP-tagged chromatin proteins. In our modified scheme called "nanoCUT&RUN", the nuclease recruitment is accomplished by the binding of a GFP-specific nanobody [24], similarly expressed as a fusion protein, to a GFP-tagged target protein produced *in vivo*. Using nanoCUT&RUN, we were able to profile the binding of the known TRL transcription factor. We were also able to reveal the distribution of telomere capping proteins on retro-elements from Drosophila telomeres and unexpectedly at some of the centromeric regions.

## Results and discussion

### Designing the nanoCUT&RUN method

In the original ChIC and CUT&RUN schemes, a bacterially expressed Protein A (ProA) fused to MNase was used to tether the nuclease at antibody-bound sites of specific chromatin proteins [16,17]. In our nanoCUT&RUN design, a single domain antibody recognizing the GFP motif replaces the ProA moiety (Fig 1A). We chose the Vhh4 clone of nanobody, which has been widely used in studies as a way to accomplish specific protein-protein interactions *in vivo* [24,25].

We therefore expressed and purified from bacteria a new fusion protein in which MNase was fused to the GFP nanobody (nGFPMNase). As shown in S1A Fig, we were able to achieve a reasonable purification of this reagent. We tested the function of this fusion protein in the following two ways. First, we showed that nGFPMNase binds GFP *in vitro*. This was done by loading non-denaturing gels with protein samples that contain both GFP and nGFPMNase and visualizing the running position of GFP under a UV light. As shown in S1B Fig, the combination of GFP and nGFPMNase retards the migration of GFP, indicating binding between the two proteins. A 1:1 molar ratio of GFP and nGFPMNase was sufficient to retard most if not all of the GFP molecules on a native gel (S1B Fig). We next tested the ability of nGFPMNase to digest DNA, and importantly whether such nuclease activity can be activated by the presence of calcium similar to the activity of the original MNase. We mixed purified plasmid DNA with nGFPMNase with or without calcium (S1C Fig). Plasmid DNA was digested to completion in the presence of calcium demonstrating that the nuclease activity is

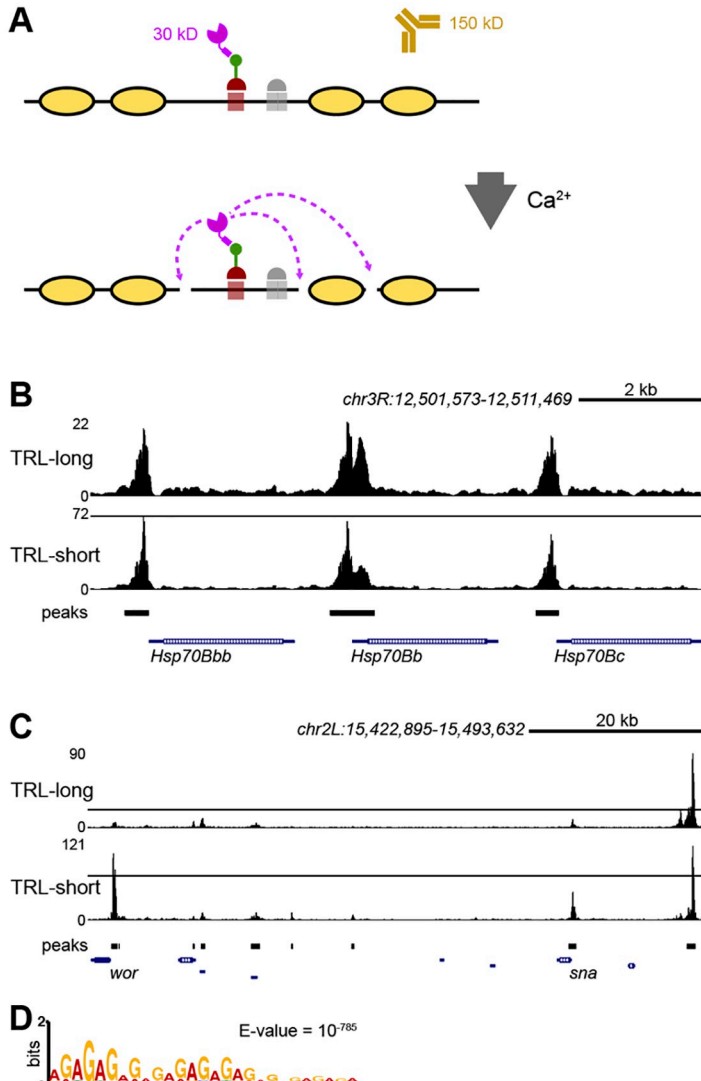

**Fig 1. Chromatin profiling of TRL by nanoCUT&RUN. A**: schematic of the method. Relative size of the nGFPMNase (magenta) to an antibody (brown) is shown. nGFPMNase binds to the GFP (green) tag of the protein of interest (red). In the presence of $Ca^{2+}$, MNase digests DNA (black line) that is not protected by the nucleosomes (yellow). **B**: landscapes of TRL isoform binding at three Hsp70 promoters. Genomic coordinates (in nt) and the scale of the *hsp70* region on *3R* are shown at the top. **C**: landscapes of preferential binding of TRL-short at the promoters of the *worniu* (*wor*) and *snail* (*sna*) genes, while an intergenic site binds both TRL isoforms (shown at the right end of the profile). Genomic coordinates and the scale of the region are shown at the top. **D**: consensus motifs of TRL-short sites.

enhanced by calcium. Therefore, our bacterially produced nGFPMNase effectively binds GFP and cleaves DNA, providing a suitable reagent for CUT&RUN profiling.

## Chromatin profiling TRL isoforms with nanoCUT&RUN

As a proof of principle for nanoCUT&RUN, we chose the well-characterized GAGA factor encoded by the *Trithorax-like* (*Trl*) gene in Drosophila [26]. We used two transgenic lines generated by the modERN project in which TRL is epitope-tagged at its C-terminus [flybase.net]. In both lines, tagged TRL proteins are expressed from a Bac transgene carrying endogenous regulatory elements of *trl*. The '804' line produces a tagged 519 aa 'TRL-short' isoform, while

the '811' line produces a tagged 567 and 611 aa 'TRL-long' isoforms. Both long and short isoforms carry a zinc-finger DNA binding domain and a BTB/POZ homodimerization domain, but differ by the length of a poly-glutamine-rich segment, which serves as the transcriptional activator [27]. TRL-short is expressed continuously, while expression of TRL-long begins in mid-embryogenesis [28].

We collected 0–12 hour old embryos, isolated nuclei, and performed nanoCUT&RUN profiling on duplicate samples, including samples lacking any GFP tag as a specificity control (no-tag control). About 12–22 million paired end reads were sequenced for each sample and mapped to the dm6 assembly of the Drosophila genome (S1 Table). Replicate chromatin profiles for TRL-short were highly correlated (Spearman's correlation 0.9), while those for TRL-long had a more moderate correlation (Spearman's correlation 0.66). Peak calling by MACS2 identified 8,332 for the TRL-long isoform (S2 Table), and the TRL-short isoform was also present at these sites. Peak calling of TRL-short isoform profiles identified substantially more (11,438) binding sites (S3 Table), suggesting that some sites preferentially bind the TRL-short isoform. Indeed, differential peak analysis identified 3,663 peaks significantly enriched for the TRL-short isoform relative to TRL-long binding (S4 Table). About 48% of TRL-short and 57% of TRL-long peaks overlap with sites identified by ChIP-seq [29] (S2 Fig), confirming that nanoCUT&RUN detects *bone fide* TRL binding sites. TRL binding was previously described at the promoters of the *Hsp70* genes [30], and both TRL-short and TRL-long isoforms coincide at these promoters (Fig 1B). In contrast, many TRL-short-enriched peaks fall at the promoters of developmentally regulated genes that are expressed in embryos, such as *N*, *wg*, *aop*, *sog*, *hid*, *wor*, and *sna* (Fig 1C). TRL-short is more abundant than TRL-long in the early embryo [28], but other neighboring peaks show similar relative magnitudes for both isoforms, again implying that many sites preferentially bind TRL-short. While differential analysis scored 837 sites as specifically enriched for TRL-long, almost all of these entries fall in highly repetitive unmapped contigs (S4 Table). We did not consider these sites further.

The two TRL isoforms both contain a common zinc-finger DNA binding domain that recognizes a 'GAGA' motif [31], so why does TRL-short preferentially bind some sites? The consensus motif for TRL-short enriched sites is an extended 'GA' repeat (Fig 1D), consistent with oligomer binding of TRL [32,33], and these are found precisely at many sites of TRL-short signal. In contrast, sites enriched for TRL-long tend to fall in heterochromatic regions and in transposon repeats, and show extended smears of signal across repetitive sequences. Motif analysis of these regions is dominated by common repeat sequences, but embedded consensus GA motifs are present. It is intriguing that TRL shifts from euchromatic binding sites to heterochromatic sites during mitosis [34], perhaps related to the preference of TRL-long for a distinct set of sequence contexts. These differences highlight the utility of epitope-tagging protein isoforms for chromatin profiling where isoform specific antibodies may not be available.

## Profiling of telomere capping proteins with nanoCUT&RUN

We were encouraged by our initial success in profiling TRL with nanoCUT&RUN and proceeded to apply the method to telomeric factors that we have been studying. The HOAP and HipHop proteins function as a complex that is specifically enriched at all telomeres in Drosophila [5,6]. Previously using a single telomere with defined sequences from the *white* gene, we showed that HipHop and HOAP, along with their interacting HP1 protein occupy a large telomeric domain from the very end of the chromosome [6]. This suggests that these proteins maintain a specialized chromatin structure at telomeres.

Although our prior study provided the first detailed picture of how these important capping proteins are distributed on telomeres, it nevertheless suffers two drawbacks. First, the previous

results were derived from a traditional ChIP plus qPCR assay in which a limited number of primer pairs from the telomeric region (about 1 kb apart and covering 11 kb in total) were used, thus greatly limiting the resolution. Second, the natural binding sites of these proteins are telomeric transposons. It remains possible that they distribute differently at native telomeres. Furthermore, the specific enrichment of these proteins at telomeres was established based on immunostaining results. It is possible that they have minor but important localization at non-telomeric positions. Therefore, we set out to use nanoCUT&RUN to profile binding of HipHop and HOAP, taking advantage of two fly lines in which the proteins of interest are expressed with a GFP tag. For HOAP, we used a knock-in line in which the endogenous *cav* locus was tagged [19,35]. For HipHop, we constructed a transgene in which the *hiphop* locus was tagged at the N terminus, and expressed from its endogenous regulatory elements. We showed that this transgene is able to rescue early larval lethality of a *hiphop* deletion mutation previously generated [35] suggesting that the GFP-HipHop protein is functional. Similar to profiling of TRL we used 0–12 hr old embryos. We performed nanoCUT&RUN profiling on GFP-HipHop, GFP-HOAP and a no-tag control with two different digestion durations (2 and 15 mins). The different digestions yielded highly similar results (Wild type Spearman's rho = 0.94, P<10–16; HOAP is Spearman's rho = 0.92, P<10–16; and HipHop Spearman's rho = 0.95, P<10–16), therefore we generated two additional biological replicates for each protein with a 2-minute digestion duration. About 25–36 million paired end reads were sequenced for each sample and mapped to the heterochromatin-enriched *D. melanogaster* genome assembly [36].

## HipHop and HOAP binding sites are enriched with telomeric elements

When reads from nanoCUT&RUN were plotted on genome assembly zoomed in on telomeres, we clearly see an enrichment of both GFP-HipHop and GFP-HOAP at the telomeric elements. For each protein, nanoCUT&RUN profiling between the replicates generated consistent results (S3 Fig and S5 and S6 Tables). Fig 2 shows such a zoomed-in view of the telomeric region from chromosome *3R*. Similar enrichment was observed for telomeres from all major chromosomes (S4 Fig, results are consistent between 2- and 15-minute digestion; S5 Fig). Because the fly strains may differ in their organization and sequence of telomeric regions, and we mapped to a reference genome assembly, we sequenced a genomic DNA control for each fly strain expressing GFP-HipHop or GFP-HOAP, and the no-tag control. The read distribution is shown in S6 Fig. We do see some variation in genomic background between the strains suggesting that differences between HOAP and HipHop on telomere 3L are driven by structural variation (S6B Fig). However, the overall pattern of HipHop and HOAP enrichment at the telomeres is the same. When we specifically examined the top 20% most abundant and enriched transposable elements presented in HipHop and HOAP nanoCUT&RUN reads relative to the no-tag control samples, the three classes of telomeric elements [*HeT-A*, *TART* and *TAHRE* (*HTT*)] are the most enriched elements (S7 Fig and S7 Table). Besides transposons, some simple repeats are also enriched in HipHop and HOAP reads (S8 Table).

## Similar distributions of HipHop and HOAP over telomeric elements

To uncover any preferred binding sites along the telomeric elements for the two proteins, we piled nanoCUT&RUN reads from either HipHop or HOAP on consensus retrotransposons from each of the three classes (*HeT-A*, *TART* and *TAHRE*) that we built based on previous annotations [12,37]. As shown in Figs 3, S8 and S9, HipHop reads are distributed along the entire lengths of all three elements with a possible exception at a region of about 6 kb in size in *TART-A* (Fig 3D). This region lies in the 5' UTR of *TART-A* [38], just upstream of *orf1* that

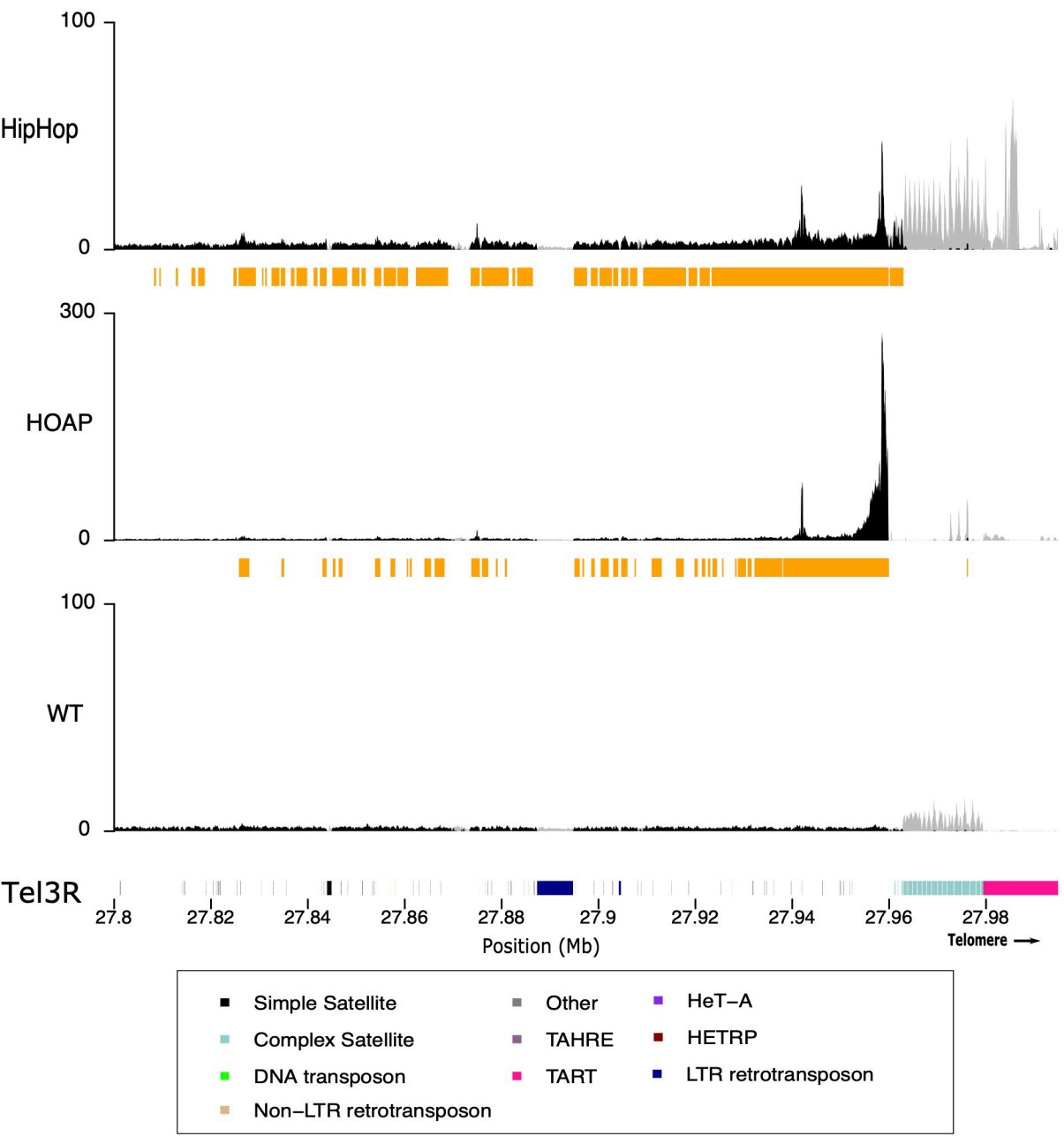

**Fig 2. Distribution of HipHop and HOAP on the telomere of chromosome *3R*.** The y-axis represents the normalized enrichment of the target protein (HipHop or HOAP) or the no-tag control (WT) for replicate 1 in RPM. The gray lines correspond to multi-mapped reads, the black lines correspond to the uniquely mapped reads. The orange bars below each plot correspond to MACS2 peaks based on the uniquely mapping reads. The colored cytoband at the bottom of the plot shows the repeat organization. The color code is shown in the legend. The distribution of the two proteins on other telomeres are shown in S4 and S5 Figs.

encodes the Gag protein. As shown in S9H Fig, this might be related to the fact that most of the *TART* elements are 5' truncated in this genetic background. Similarly, we observed a loss of HipHop and HOAP enrichment at around 9 kb of the *TART-B* element (Fig 3C), which is due to a deletion of this region in most of the copies in this genetic background (S9G Fig). On the most abundant telomeric element, *HeT-A*, HipHop reads are more or less evenly

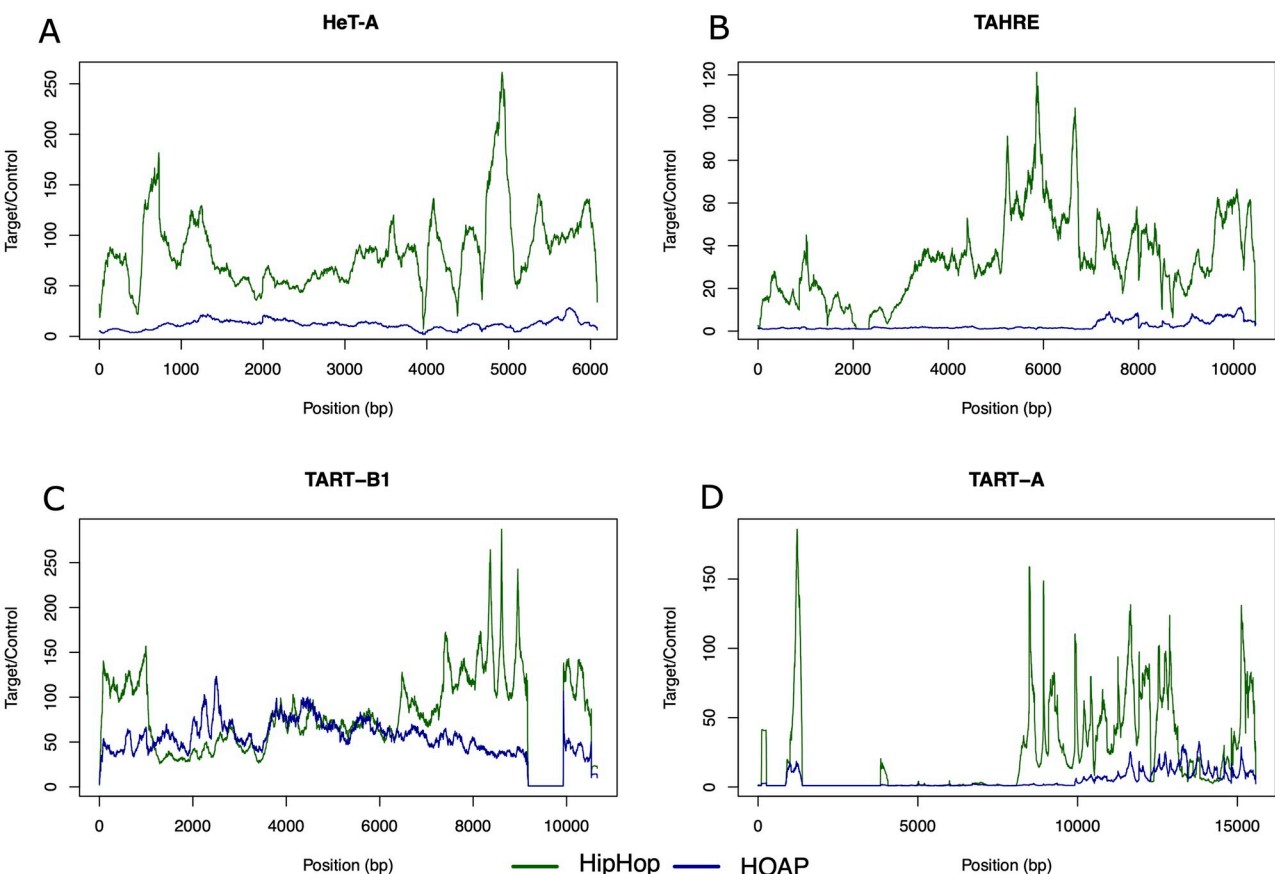

**Fig 3. HipHop and HOAP distributions on telomeric retro-elements.** Distribution of HipHop and HOAP on the consensus sequences of *TART-A*, *TART-B1*, *HeT-A*, and *TAHRE* elements. The y-axis represents the mean normalized enrichment (in RPM) of the two replicates for each target protein (HipHop or HOAP) over the no-tag control.

distributed along the entire length (Fig 3A) and this pattern is consistent among all *HeT-A* subfamilies (S8 Fig). This suggests that HipHop might not have a strongly preferred DNA sequence within each *HTT* element for binding. Consistently, we did not uncover any particularly strong motif(s) enriched among HipHop reads under the high confidence (irreproducible discovery rate<0.05; S5 Table; see Materials and methods) peaks.

The distribution of HOAP reads generally tracks those of HipHop. This pattern is also evident looking at the distribution of HipHop and HOAP along the parts of the telomere represented in our genome assembly (Figs 2 and S4–S6). The enrichment extends beyond the HTT domain and into the sub-telomeric region >100 kb from the distal end of the chromosome assembly. HOAP is generally less enriched than HipHop at telomeric repeats (Figs 3 and S8), with the exception of *TART-B* and *Het-A5*. We do not know if this reduction of HOAP occupancy is related to the two genes having different expression levels. The exception for TART-B and Het-A5 may be driven by variation in HOAP enrichment within or between telomeres, although we cannot exclude the possibility that HOAP has a sequence preference for these elements. Interestingly, along the consensus sequence of *TART-B*, HOAP appears as enriched as HipHop (Fig 3C). In addition, on the consensus *TAHRE* element, significant HOAP enrichment is limited to the very 3' end (Fig 3B). Given the fact that *TAHRE* is the least abundant of the three retrotransposons [38–40], it is possible that *TAHRE* polymorphisms among different strains alone could account for this observation, as most *TAHRE* elements are 5' truncated in

the *cav^{gfp}* strain (S9F Fig). Therefore, we conclude that HipHop and HOAP share similar distribution patterns at the telomeres, and that they bind indiscriminately along the HTT elements without preferred binding sites.

## Enrichment of telomeric proteins at centric heterochromatin

Unexpectedly, we detected an enrichment of telomere proteins on islands of repeats that correspond to the centromeres (S3, S7 and S10 Figs). The primary enrichment is of HipHop on the 4th and *X* chromosome centromere (Fig 4). This pattern does not seem to be driven by any particular sequence within the centromere. Prior immunostaining studies localizing HipHop in relationship to telomeres were performed on (1) polytene cells from third instar larvae, or (2) mitotic cells with condensed chromosomes [6,35]. The centromeric regions are under-replicated in polytene cells [41,42]. Mitotic centromeres might have poor accessibility to antibodies making it difficult to detect weaker signals than those at telomeres. These factors might have prevented us from detecting centromeric HipHop cytologically.

The *X* and 4th chromosomes are both acrocentric, having a very short arm so that centric heterochromatin is relatively close to a telomere. Whether this common feature leads to the enrichment of HipHop at their centromeres in particular requires further investigations. One could imagine that the spreading of HipHop-enriched telomeric chromatin might encroach the centromeres of acrocentric chromosomes. However, we consider "spreading" an unlikely mechanism for the appearance of HipHop on centromeric regions of acrocentric chromosomes. Although the centromeres and telomeres of chromosomes *X* and *4* appear "close" in cytological images, the physical distances are in the megabases range for the *X* chromosome. It

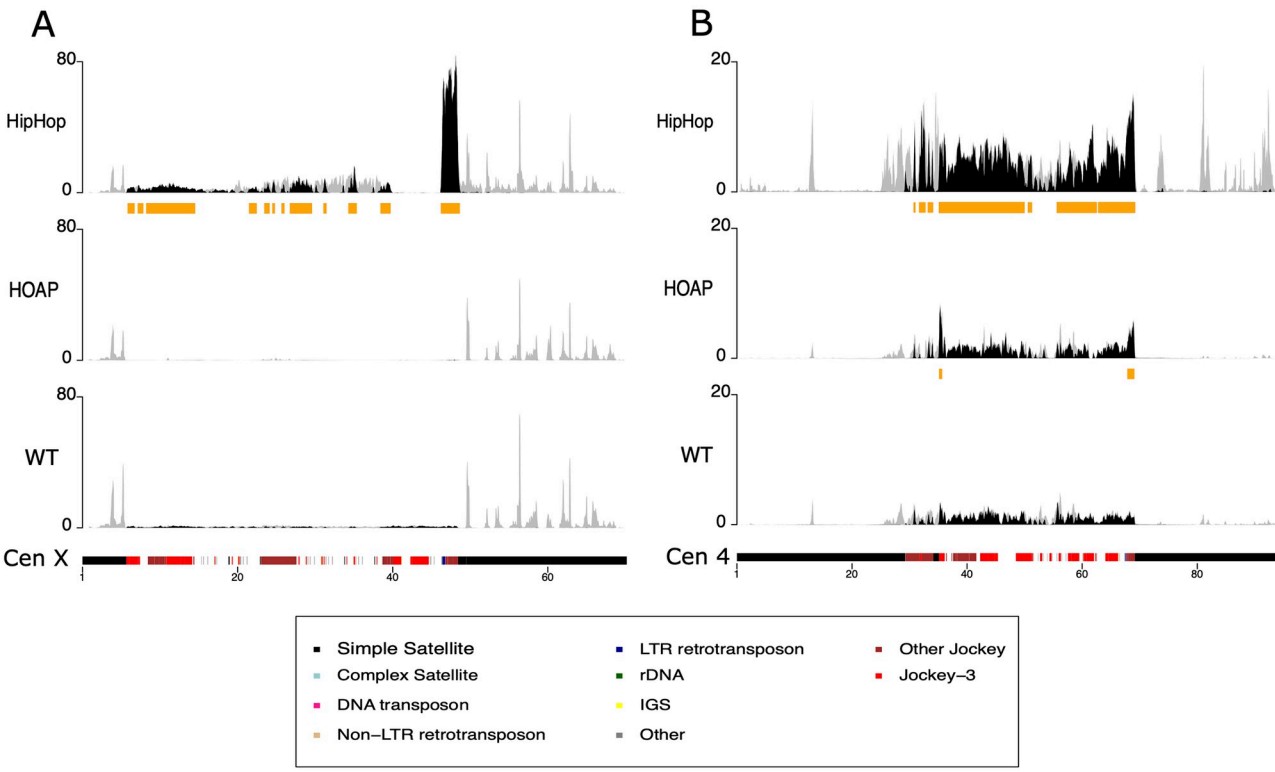

**Fig 4. Distributions of HipHop and HOAP on centromeres of the *X* and 4th chromosomes.** The y-axis represents the normalized enrichment of target protein (HipHop or HOAP) or the no-tag control (WT) for replicate 1 in RPM. The gray lines correspond to multi-mapping reads, the black lines correspond to the uniquely mapping reads. The orange bars below each plot correspond to MACS2 peaks based on the uniquely mapping reads. The colored cytoband at the bottom of the plot shows the repeat organization. The color code is shown in the legend.

is difficult to envision that HipHop-enriched chromatin could spread for thousands of kilo-bases from the end of the *X* chromosome.

Interestingly, our recent results showed that a hypomorphic *hiphop* mutation behaves as a recessive suppressor of heterochromatin-induced Position-Effect-Variation (PEV) [43]. More-over, the specific case of PEV used in that study involves the *X* centric heterochromatin. These earlier results seem to be consistent with the presence of HipHop in centric heterochromatin as revealed by this study, and with a potential role of HipHop in silencing that is not limited to chromosome ends.

## Concluding remarks

Here we developed the nanoCUT&RUN technique that could be a powerful addition to the series of improvements/extensions to the original CUT&RUN design. An advantage of nano-CUT&RUN is that it is readily applicable to any protein tagged with GFP. In model organisms with facile genetics, the normal function of the GFP-tagged proteins can be routinely verified by testing their effects on rescuing mutant phenotypes, thus providing additional confidence for the nanoCUT&RUN data. In addition, the nanoCUT&RUN method is advantageous when one's goal involves comparing different target proteins or isoforms of a single target, since all profiling using our method is based on the same nanobody-GFP interaction. While we were preparing our manuscript, Koidl and Timmers [44] reported the "greenCut&Run" method in mammalian cells, which is based on an identical principle as nanoCUT&RUN.

We confirmed the feasibility of this approach using the well-studied TRL transcription fac-tor, and demonstrated a useful application of this approach by profiling two telomere capping proteins in Drosophila for the first time. Our results confirm that telomeric capping in Dro-sophila is unlikely to require a specific DNA sequence at chromosome ends. In addition, we uncovered an enrichment of HipHop at centromeric regions, which seems consistent with prior phenotypic analyses of *hiphop* mutants.

## Materials and methods

### Drosophila stocks and genetics

Drosophila stocks were raised on standard cornmeal-based food and kept at a 25°C incubator with constant lighting. Two transgenic lines: BL64804 (flybase genotype: w[1118]; P{y[+7.7] w [+mC] = Trl.BCDEH-GFP.FPTB}attP40) and BL64811 (flybase genotype: y[1] w[*]; P{y[+7.7] w[+mC] = Trl.IJ-GFP.FPTB}attP40) were obtained from the Bloomington Drosophila Stock Center. Each carries an insertion of a BAC clone in which one of the isoforms of the TRL pro-tein is tagged with GFP (flybase.net). The *cav*$^{gfp}$ allele encoding GFP-tagged HOAP proteins was described and characterized [19,35], and it was used in nanoCUT&RUN profiling of HOAP. The *hiphop*$^{L41}$ deletion allele of the *hiphop* gene was described [35]. To generate a *gfp*-tagged *hiphop* transgene, a *hiphop* genomic clone previously used [35] was modified by insert-ing a *gfp* gene at the N-terminus of *hiphop* by recombineering [45]. This clone was inserted into the genome at the *attP* site carried by the *P{CaryP}attP40* element at position of 25C by phiC31 mediated site-specific integration. A stock that is homozygous for both the *hiphop*$^{gfp}$ transgene on chromosome *2* and the *hiphop*$^{L41}$ mutation on chromosome *3* was used in nano-CUT&RUN profiling of HipHop. The *w*$^{1118}$ stock was used as the no-tag control.

### Embryo collection and nanoCUT&RUN

Drosophila strains were cultured at 25°C on corn meal medium. Overnight (0-12h old) embryos were collected. They were washed off of grape juice-agar plates with Embryo Washing

Buffer (EWB, 0.7% NaCl, 0.04% Triton-X100), dechorionated with 50% bleach, washed twice with EWB, and stored at -80˚C before use. To purify nuclei from embryos, about 30μl of embryos were suspended in 500μl of Buffer B (pH7.5, 15mM Tris-HCl, 15mM NaCl, 60mM KCl, 0.34M Sucrose, 0.5mM Spermidine, 0.1% β-mercaptoethanol, 0.25mM PMSF, 2mM EDTA, 0.5mM EGTA), and grinded with a pestle on ice. The grinded mixture was transferred to a 1.5ml Eppendorf tube and spun for 5min at 5000G and 4˚C. The pellet was resuspended and washed with 500μl of Buffer A (pH7.5, 15mM Tris-HCl, 15mM NaCl, 60mM KCl, 0.34M Sucrose, 0.5mM Spermidine, 0.1% β-mercaptoethanol, 0.25mM PMSF). The wash was repeated twice, and the nuclei were resuspended in 600μl of WBSED buffer (20 mM HEPES pH 7.5, 150 mM NaCl, 0.1% BSA, 2 mM EDTA, 0.5mM Spermidine, 0.05% digitonin, 1X complete EDTA-free protease inhibitor from Roche).

NanoCUT&RUN was performed based a protocol for CUT&RUN (dx.doi.org/10.17504/protocols.io.zcpf2vn) with the following modification: a 30 μL volume of starting embryos resulted in 600 μL of nuclei suspended in WBSED to which 0.6 μL of a 0.4 mg/mL stock of nGFPMNase recombinant protein was added. The MNase released DNA was suspended in 20 μL of dH2O. A detailed nanoCUT&RUN protocol is provided in S1 Protocol.

## Purification and characterization of the nGFPMNase protein

The coding regions for the MNase nuclease domain and an anti-GFP nanobody were codon-optimized for expression in *E.coli* and synthesized by IGE Biotech (Guangzhou, China). They were cloned into the pET28a vector so that nGFPMNase has a N-terminal His6 tag. A map is included in S1D Fig, and sequence of the plasmid is available upon request. Bacterial expression and purification were performed using standard protocols with an IPTG concentration of 0.5mM for induction and an Imidazole concentration of 250mM for elution. A detailed purification protocol is provided in S2 Protocol.

## Sequencing and data analyses

Libraries were sequenced with 150 paired-end mode on the Illumina Hiseq X10/Nova seq platform by AceGen (Guangzhou, China). Sequencing data have been submitted to the NCBI short reads archive with the accession number PRJNA723550 (S1 Table).

## Peak calling

For telomere protein profiling, we trimmed the reads with Trim Galore (https://github.com/FelixKrueger/TrimGalore/) (paired end default settings), and then mapped the trimmed reads to a heterochromatin-enriched genome assembly [36] using bowtie2 [46]. We defined uniquely mapped reads using samtools (v1.5) (-q 10). We called peaks using MACS version 2.1.1.20160309 [47] (-q 0.01; hereafter referred to as MACS2 peaks).

For TRL profiling, we trimmed paired-end reads with Trim Galore within Galaxy (usegalaxy.org) with default settings except hard-clipping 3 bp off 3' ends of reads. Trimmed reads were mapped to the dm6 assembly with bowtie2 (-I 20 -X 1000, mate dovetailing, one mate alignment to contain another, very sensitive end-to-end). We called peaks using MACS2 Galaxy version 2.1.1.20160309.6 and differential enrichment of peaks between TRL-long and TRL-short datasets using DiffBind Galaxy version 2.10.0. Peak calls are provided in S2 and S3 Tables.

## Irreproducible discovery rate analyses

We performed an irreproducible discovery rate (IDR) analysis (https://github.com/nboley/idr) to identify high confidence peaks that overlap between replicates (IDR<0.05, represented by

black dot in S3A and S3C Fig). We considered 2-min and 15-min samples as replicates and ran the IDR analysis on the MACS2 peaks. The localization of those high confidence peaks (S3B and S3D Fig) confirmed that the majority of the telomere proteins are localized in telomeric regions. We further used STREME (https://meme-suite.org/meme/tools/streme) to perform motif analysis with the fragment sequences under the high confident peaks (IDR<0.05) of HipHop and HOAP (no-tag samples as control, and 2-min and 15-min samples as replicates), and discovered no specific motif enriched for HipHop or HOAP.

### Repeat analysis

We performed analyses to determine the repeats enriched for HipHop and HOAP. For complex repeats (e.g., complex satellite DNAs with repeat unit > 100bp, TEs), we mapped trimmed reads to a heterochromatin-enriched genome assembly [36] using bowtie2 [46] (default settings), and summarized read counts for each complex repeat using custom python scripts. We normalized read count to the number of mapped reads and report RPM (Reads Per Million). We calculated enrichment values as IP(RPM)/control(RPM), and considered a repeat to be enriched only if it is in the top 20% of the IP RPM and also top 20% of the IP/control enrichment. For the HTT elements, we analyzed each subfamily [12] separately (S7 Table). Because the enrichment for subfamilies of *HeT-A* elements were similar, we combined onto a single consensus (Figs 3 and S7). For *TART* elements, *TART-A* and *TART-B* show different enrichment patterns, therefore we show both subfamilies (Figs 3 and S7). We also calculated the enrichment for the centromere islands as described previously (S3 Fig) [37]. To determine which parts of HTT are represented in the enrichment, we examined the read pileup patterns along their consensus sequences. We used BLAST (v2.6.0) to map either reads matching HTT or genomic HTT variants (as a control) to the consensus dimer of the HTT, and then converted coordinates along a dimer to coordinates along a monomer consensus sequence.

For simple tandem repeats, we summarized overrepresented k-mers in the trimmed reads using kseek [48; https://github.com/weikevinhc/k-seek], and normalized the k-mer count to the number of mapped reads to the assembly and report the RPM value. We calculated the enrichment values as IP(RPM)/control(RPM), and considered the k-mers to be enriched if RPM>10 and enrichment value >1 in both replicates. Due to the repetitive nature of the elements enriched at telomeres, we used all the mapped reads including reads that have multiple mapping locations in the genome.

Data underlying figures are deposited in the Dryad Digital Repository [49].

### Dryad DOI

https://doi.org/10.5061/dryad.zcrjdfngc [49].

### Supporting information

**S1 Fig. Characterization of nGFPMNase. A**: purification of the nGFPMNase fusion protein from bacteria. Extracts from different fractions were run on SDS-Page and stained with Coomassie Blue. Lanes 1: purified nGFPMNase; 2: insoluble fraction from bacteria overexpressing nGFPMNase; 3: soluble fraction; 4: total extract from overexpressing bacteria; 5: total extract from uninduced bacterial culture. "M" denotes protein markers with sizes in KD indicated to the right. The arrow marks the running position of nGFPMNase. **B**: nGFPMNase (nanoGM) binds GFP. GFP fluorescence from a native protein gel is shown with protein components loaded onto each lane shown at the top. Note that nGFPMNase alone does not emit fluorescence. The double star marks the running position of the complex between nGFPMNase and GFP. The single star marks the running position of GFP alone. **C**: nGFPMNase (nanoGM)

digests DNA in the presence of calcium. Plasmid DNA was mixed with purified nGFPMNase in the nuclease digestion buffer with or without calcium. "M" denotes DNA markers with sizes indicated to the right. **D**: map of the nGFPMNase expression plasmid.
(JPEG)

**S2 Fig. Comparison of TRL mapping by nanoCUT&RUN and by ChIP-seq. A**. Comparison of peak lists from embryos detected by ChIP-seq [29] and by nanoCUT&RUN for TRL. **B**. Heatmaps of nanoCUT&RUN signal on 6,373 peaks called on ChIP-seq data. The ChIP-seq peak list was published as Supplementary File 1 in [29].
(AI)

**S3 Fig. IDR analysis.** Panels **A** and **C** represent the peak scores of replicate 1 *versus* replicate 2 on a log10 scale. The IDR analyses detected 1686 peaks in common between the two HipHop replicates but only 138 peaks passed the cutoff of IDR<0.05 (in black). The IDR analyses detected 307 peaks in common between the two HOAP replicates but only 58 peaks passed the cutoff of IDR<0.05 (in black). Panels **B** and **D** represent the localization of the peaks with an IDR <0.05. The majority of those peaks are localized on telomeres. We also detect a minority of peaks on the centromeres. All the peaks localized outside the telomeres and centromeres are grouped in the category "Other". However, in this category most of the peaks actually localized on one Y-linked scaffold (Y_scaffold4), which is also enriched in HTT, however this scaffold is unlikely to represent the Y telomere because of its cytological location [37].
(PDF)

**S4 Fig. Distribution of HipHop and HOAP on the telomeres of chromosome *2R* (A), *2L* (B), *4* (C), *X* (D), and *3L* (E).** The y-axis represents the normalized enrichment of target protein or the no-tag control (WT) for replicate 1 in RPM. The gray lines correspond to multi-mapped reads, the black lines correspond to the uniquely mapped reads. The orange bars below each plot correspond to MACS2 peaks based on the uniquely mapping reads. The colored cytoband at the bottom of the plot shows the repeat organization. The color code is shown in the legend.
(PDF)

**S5 Fig. Distribution of HipHop and HOAP on the telomeres of all chromosomes after 2 and 15-minute digestion.** The y-axis represents the normalized enrichment of target protein or the no-tag control (WT) for two different digestion durations (2 or 15 min) in RPM. The gray lines correspond to multi-mapped reads, the black lines correspond to the uniquely mapped reads. The orange bars below each plot correspond to MACS2 peaks based on the uniquely mapping reads. The colored cytoband at the bottom of the plot shows the repeat organization. The color code is shown in the legend.
(PDF)

**S6 Fig. Distribution of HipHop and HOAP and genomic DNA on all telomeres.** The y-axis of the first plot (white background) represents the normalized enrichment (in RPM) for a second replicate of target protein (HipHop Rep2 and HOAP Rep2) or the no-tag control (WT Rep2). The y-axis of the second plot (yellow background) represents the normalized enrichment (in RPM) of the genomic DNA coverage of each strain (WG: whole genome). The gray lines correspond to multi-mapped reads, the black lines correspond to the uniquely mapped reads. The orange bars below each first plot correspond to MACS2 peaks based on the uniquely mapping reads. The colored cytoband at the bottom of the plot shows the repeat organization. The color code is shown in the legend.
(PDF)

**S7 Fig. Enrichment of HipHop and HOAP on complex repeats.** The plot shows the normalized enrichment of target protein over the no-tag control (in RPM) for the top 10 repeats enriched in both HipHop and HOAP nanoCut&Run profiling. The full dataset is in S7 Table. (PDF)

**S8 Fig. HipHop and HOAP distributions on *HeT-A* subfamilies.** Distribution of HipHop and HOAP on the individual subfamilies of *HeT-A* from [12]. The y-axis represents the mean normalized enrichment (in RPM) of the two replicates for each target protein (HipHop or HOAP) over the no-tag control. (PDF)

**S9 Fig. Genomic pileup on HTT elements.** Genomic read coverage on HTT elements (*TART-A*, *TART-B*, *TAHRE*, *Het-A*, *Het-A 1D*, *Het-A 2*, *Het-A 3*, *Het-A 5*) of GFP-HipHop and GFP-HOAP strains. The y-axis represents the normalized reads coverage in RPM. (PDF)

**S10 Fig. Distributions of HipHop and HOAP and genomic DNA on all centromeres.** The y-axis of the first plot (white background) represents the normalized enrichment (in RPM) for a second replicate of target protein (HipHop Rep2 and HOAP Rep2) or the no-tag control (WT Rep2). The y-axis of the second plot (yellow background) represents the normalized enrichment (in RPM) of the genomic DNA coverage of each strain (WG: whole genome). The gray lines correspond to multi-mapped reads, the black lines correspond to the uniquely mapped reads. The orange bars below each first plot correspond to MACS2 peaks based on the uniquely mapping reads. The colored cytoband at the bottom of the plot shows the repeat organization. The color code is shown in the legend. (PDF)

**S1 Table. Summary of samples metrics.** Summarized the total number of reads, the number of mapping reads and the accession number for each sample. (XLSX)

**S2 Table. TRL-long binding sites.** Peaks defined by MACS2 for the TRL-long-GFP isoform. (XLSX)

**S3 Table. TRL-short binding sites.** Peaks defined by MACS2 for the TRL-short-GFP isoform. (XLSX)

**S4 Table. Differentially bound TRL sites.** Differential peaks defined by DiffBind with FDR<0.05. (XLSX)

**S5 Table. IDR peaks for HipHop.** IDR output table based on MACS2 peak calling, for HipHop protein between the biological replicates. The score column represents the scaled IDR value and is used for IDR cutoff, peaks with IDR<0.05 have score>540. (CSV)

**S6 Table. IDR peaks for HOAP.** IDR output table based on MACS2 peak calling, for HOAP protein between the biological replicates. The score column represents the scaled IDR value and is used for IDR cutoff, peaks with IDR<0.05 have score>540. (CSV)

**S7 Table. Enrichment scores for individual complex repeat and transposable element families.** Normalized enrichment scores of the target protein over no-tag control (in RPM) for the

complex satellite and transposable element enriched in HipHop and HOAP.
(CSV)

**S8 Table. Enrichment scores for simple repeats.** Normalized enrichment scores of the target protein over no-tag control (in RPM) for the simple tandem repeats enriched in HipHop and HOAP.
(CSV)

**S1 Protocol. A detailed protocol for nanoCUT&RUN.**
(DOCX)

**S2 Protocol. A detailed protocol for nGFPMNase purification.**
(DOCX)

## Author Contributions

**Conceptualization:** Yikang S. Rong.

**Data curation:** Tao Chen.

**Formal analysis:** Xiaolu Wei, Cécile Courret, Kami Ahmad, Amanda M. Larracuente.

**Funding acquisition:** Xiaolu Wei, Kami Ahmad, Amanda M. Larracuente, Yikang S. Rong.

**Investigation:** Tao Chen, Min Cui, Lin Cheng, Jing Wu.

**Methodology:** Xiaolu Wei, Cécile Courret.

**Project administration:** Kami Ahmad, Amanda M. Larracuente, Yikang S. Rong.

**Resources:** Min Cui.

**Software:** Xiaolu Wei, Cécile Courret.

**Supervision:** Amanda M. Larracuente, Yikang S. Rong.

**Validation:** Tao Chen.

**Visualization:** Tao Chen, Xiaolu Wei, Cécile Courret, Kami Ahmad.

**Writing – original draft:** Kami Ahmad, Amanda M. Larracuente, Yikang S. Rong.

**Writing – review & editing:** Tao Chen, Xiaolu Wei, Cécile Courret, Kami Ahmad, Amanda M. Larracuente, Yikang S. Rong.

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
