## [Decision Letter · Decision Letter 0]

8 Jun 2022

Dear Dr Rong,

Thank you very much for submitting your Research Article entitled 'The nanoCUT&RUN technique visualizes telomeric chromatin in Drosophila' to PLOS Genetics.

The manuscript was fully evaluated at the editorial level and by independent peer reviewers. The reviewers appreciated the attention to an important topic but identified some concerns that we ask you address in a revised manuscript

We therefore ask you to modify the manuscript according to the review recommendations. Your revisions should address the specific points made by each reviewer.

In particular, we request that you address the following from Rev 2:

- provide a genome-wide assessment of your results for TRL relative to previous work, as this is key to validation of the method.

- provide a detailed protocol for expressing and purifying nGFPMNase.

- provide the sequence of the above plasmid and deposit it at Addgene; this will greatly increase the impact of your work.

[LINK]

Yours sincerely,

Daniel A Barbash

Guest Editor

PLOS Genetics

Chengqi YI

Section Editor: Methods

PLOS Genetics

We request that you address the reviewers' comments, in particular the following from Rev 2:

- provide a genome-wide assessment of your results for TRL relative to previous work, as this is key to validation of the method.

- provide a detailed protocol for expressing and purifying nGFPMNase.

- provide the sequence of the above plasmid and deposit it at Addgene; this will greatly increase the impact of your work.

Reviewer's Responses to Questions

**Comments to the Authors:**

Reviewer #1: The revised version of Chen et al. addresses the majority of reviewer question and suggestions.

Some minor comments:

Line 61: Consider adding, “unlike the sequence-specific binding of Shelterin components in mammals, Drosophila Terminin proteins are sequence-independent."

Line 112: Consider defining “nanobody”

Line 162: How many peaks were previously called? Stating “All of these profiles show peaks” makes it unclear where there are three peaks or 3,000 peaks that are recapitulated by the TRL-S and TRL-L profiling.

Line 234: map -> mapped

Line 271: While expression certainly could explain the heterogeneity, could elevated read number for HOAP at TART-B and Het-A5 could also be related to sequence specificity?

I am still unable to read the legend of Figure 4 – please make the boxes larger.

Reviewer #2: This manuscript describes an improvement to the CUT&RUN technique for profiling GFP-tagged proteins. It then applies this approach to generate genome-wide DNA binding profiles for TRL and two telomere binding proteins, HipHop and HOAP. TRL serves as a positive control/proof-of-principle experiment because it has been profiled by similar approaches such as ChIP-seq in the past. By contrast, the HipHop and HOAP profiles represent new datasets for the field. Thus, the paper provides both technical and biological advancements. Regarding the technical advance, nanoCUT&RUN will be useful for profiling GFP-tagged proteins by employing nanobodies that recognize GFP; this represents an advance over existing methods that employ GFP antibodies, which often exhibit lot-to-lot variability (in the case of polyclonals) or low efficacy (in the case of monoclonals). Regarding the biological advance, telomere binding profiles generated here are consistent with prior evidence based on limited ChIP datapoints that HipHop and HOAP bind telomeres in a sequence-independent manner; this is an important finding for the field. Moreover, the others find unexpected binding of HipHop to the chrX and chr4 centromeres, thereby extending understanding in the field, and potentially explaining observations that HipHop mutants suppress PEV of centromere-inserted elements.

Primary comments:

The authors use TRL profiling as proof-of-principle to validate nanoCUT&RUN as a technique for profiling DNA-binding proteins. Therefore, a direct comparison of nanoCUT&RUN profiles with published TRL ChIP-seq or CUT&RUN datasets is needed. The authors show individual TRL binding events in Fig 1b; however, genome-wide comparisons are required.

The utility of nanoCUT&RUN to the scientific community will be maximized by sharing the materials and protocols required for generating nGFPMNase.

o Currently, the authors include a graphical map of the vector used to produce recombinant protein. It is necessary that the sequence of this vector be published as part of the manuscript, rather than being made available upon request. Ideally, the vector itself would be deposited where it can be acquired by the community, such as the Addgene vector database.

o Currently, the protocol used for inducing and purifying nGFPMNase is not described in any detail. It is necessary that all possible detail be included so that others can replicate their approach. The technical advance is a major part of this manuscript’s significance; other labs should be able to employ nanoCUT&RUN after reading this paper.

High-throughput sequencing approaches to studying telomere biology are hampered by highly repetitive DNA sequences. Although the new HipHop and HOAP findings are primarily descriptive in nature, they suggest that HipHop and HOAP do not bind telomeres in a sequence specific manner because the authors find no evidence of a motif and the DNA binding profiles are broad rather than peak-like (as is typically observed for transcription factors). The authors should plot their own nanoCUT&RUN TRL data at telomeric sequences; this could potentially provide a contrasting DNA binding pattern (ie. Peak-like, or absent), which would considerably strengthen the conclusion that HipHop and HOAP bind broadly across telomeres and mitigate concerns that their DNA binding pattern is due to technical rather than biological reasons.

**Have all data underlying the figures and results presented in the manuscript been provided?**

Reviewer #1: Yes

Reviewer #2: **No: **DNA sequence of plasmid encoding nGFPMNase; protocol for expressing and purifying nGFPMNase.

PLOS authors have the option to publish the peer review history of their article (what does this mean?). If published, this will include your full peer review and any attached files.

Reviewer #1: No

Reviewer #2: No

---

## [Editor Report · Decision Letter 1]

21 Jul 2022

Dear Dr Rong,

We are pleased to inform you that your manuscript entitled "The nanoCUT&RUN technique visualizes telomeric chromatin in Drosophila" has been editorially accepted for publication in PLOS Genetics. Congratulations!

Please modify the Data Availability section and the manuscript to indicate that the expression plasmid is available from Addgene rather that 'by request'.

Yours sincerely,

Daniel A Barbash

Guest Editor

PLOS Genetics

Chengqi YI

Guest Editor

PLOS Genetics

Comments from the reviewers (if applicable):

Please modify the Data Availability section and the manuscript to indicate that the expression plasmid is available from Addgene rather that 'by request'.

**Data Deposition**

http://datadryad.org/submit?journalID=pgenetics&manu=PGENETICS-D-22-00479R1

**Press Queries**

---

## [Editor Report · Acceptance letter]

27 Aug 2022

PGENETICS-D-22-00479R1 

The nanoCUT&RUN technique visualizes telomeric chromatin in Drosophila 

Dear Dr Rong, 

We are pleased to inform you that your manuscript entitled "The nanoCUT&RUN technique visualizes telomeric chromatin in Drosophila" has been formally accepted for publication in PLOS Genetics! Your manuscript is now with our production department and you will be notified of the publication date in due course.

With kind regards,

Zsofia Freund

PLOS Genetics

On behalf of:
